# Implementation research protocol on the national community health policy in Guinea: A sequential mixed-methods study using a decision space approach

Alexandre Delamou[1,2], Fassou Mathias Grovogui[1,2]*, Lior Miller[3], Amy Nye[3], Mamadi Kourouma[4], Delphin Kolié[1,2], Tohanizé Goumou[4], Thomas J. Bossert[5]

**1** African Centre for Excellence (CEA-PCMT), University Gamal Abdel Nasser of Conakry, Conakry, Guinea, **2** Maferinyah Training and Research Center in Rural Health, Forécariah, Guinea, **3** Results for Development Institute, Washington, D.C., United States of America, **4** National Directorate of Community Health and Traditional Medicine, Ministry of Health and Public Hygiene, Conakry, Guinea, **5** Harvard T.H. Chan School of Public Health, Boston, Massachusetts, United States of America

* mgrovogui@maferinyah.org

## Abstract

The overall goal of this study is to explore the rollout of the community health policy in Guinea in the context of decentralization, and the role of decision space (the decision authority, capacities, and accountability of local officials) in explaining gaps between the policy's conceptualization and actual implementation. The implementation research study will employ a sequential explanatory mixed-methods design. The study will be conducted in 27 communes purposefully selected across the country and include communes where the national community health policy is fully, partially, and not yet being implemented. The quantitative component, based on a survey questionnaire and secondary data, will use ordinary least squares (OLS) multiple regression to compare maternal and child health (MCH) coverage indicators according to the level of policy implementation in the commune. An interrupted time series analysis will be conducted to assess changes in routine MCH service delivery indicators associated with implementation of the community health policy, comparing indicators from one year prior to implementation. OLS regression will be conducted to assess the association between decision space and MCH indicators; all analyses will be carried out in Stata. Findings from the quantitative study will be used to inform the key qualitative questions and areas to explore in greater depth, to develop the interview and focus group guides, and to generate an initial codebook. Qualitative data will be double coded in NVivo by two qualitative analysts, and results generated using thematic analysis. Findings from the quantitative and qualitative components will be integrated and triangulated for interpretation and reporting. Findings and recommendations of this study will inform revisions to the National Community Health Policy to improve its rollout and effectiveness.

**Data Availability Statement:** All relevant data from this study will be made available upon study completion

**Funding:** This study was supported by the USAID through a grant (No. 7200-AA-18CA-00037).

**Competing interests:** The authors have declared that no competing or conflict interests exist.

# Introduction

## Rationale and background

There is growing international recognition and consensus that community health workers (CHWs) play a pivotal role in the functioning of health care and public health services in low- and middle-income countries (LMICs), especially in ensuring coverage of essential health services for underprivileged populations such as those living in rural and remote areas [1–3]. CHWs are also crucial for enhancing health system resiliency, given their key roles in community mobilization, service provision and community-based surveillance in the context of natural disasters, conflicts, and epidemics [1,3]. CHWs also help address the growing shortage of health care workers (HCWs) in LMICs, a significant concern for attaining global health security and universal health coverage [4, 5].

The efficacy and cost-effectiveness of CHWs programs have widely been documented in sub-Saharan Africa [6, 7]. For example, substantial increases in maternal and child health (MCH) indicators–such as antenatal (ANC) and postnatal care (PoNC) visits, immunization coverage, and exclusive breastfeeding–were achieved in South Africa two years after implementing the national CHW program [8, 9]. Modeling estimates indicate that expanding population coverage of evidence-based MCH interventions using CHWs would save up to an estimated 7 million lives globally over a five-year period [10]. The scale-up of evidence-based nutrition interventions in community and primary health care settings for children under five would reduce deaths by 15% [11]. This evidence has led to multiple initiatives and commitments from governments and donors to finance, expand, and institutionalize community health programs [12]. However, in under-resourced, uncoordinated, and poorly regulated public healthcare and health systems, constraints including attrition and absenteeism of CHWs have been reported [13, 14]. Limited ownership, participation, and decision-making capabilities and power of local communities have also been identified as challenges influencing the efficacy of CHW programs [7]. In addition, research indicates that there are certain contextual factors that must be met for CHW programs to be effective, including the existence of supportive supervision, continuous education, and adequate logistical support and supplies [15]. Notwithstanding these challenges, CHW programs have the potential to shape the achievement of equity, efficiency, and resilience of health systems [16–19]. Evidence suggests that CHW programs can be optimized by bringing decision-making and human and financial resource governance closer to local communities through decentralization accompanied by adequate institutional capacity and accountability [20, 21].

Health sector decentralization reforms have the potential to increase responsibilities of local decision-makers and provide opportunities for citizen participation in local health systems management and accountability [20, 22]. Decentralization is, however, a complex, long-term, intervention involving a variety of actors, including actors outside the control of the health sector, with divergent interests and influences [23]. Evidence from Kenya, Indonesia, South Africa, and other settings suggests that decentralization can have unintended consequences, for example pre-existing patronage systems may lead to the unequal distribution of resources and prioritization of development programs thought to contribute more to economic growth over health and social development programs [24]. Therefore, successful decentralization requires addressing pre-existing contextual norms and practices, including putting in place mechanisms that increase national actors' willingness to provide clear guidance, ensure priority-setting capacity, and stimulate community accountability and meaningful ownership and engagement [24, 25]. Further, there is a lack of evidence on the influence of decentralization on community health programs [15].

## Conceptual framework

The decision space approach, or the decision authority, capacities, and accountability of local officials, provides a useful conceptual framework to explore the rollout of community health programs, and has been applied in multiple contexts implementing decentralization reforms in the health sector [21, 26]. The decision space approach involves issues about the decision-making authority and choices that can be made by local officials, and how these interact with institutional capacities and accountability.

Decision space is defined as having two components. The first, *de jure* decision space, is the degree of choice that local, decentralized officials are authorized to make as is written in official strategies, policies, or laws [27]. The second element is *de facto* decision space, or the degree of power that local officials wield in practice. In applying this approach, research teams can better understand the gaps between the conceptualization of new policies such as community health programs, and their actual implementation, particularly in the context of decentralization.

## Guinea's national community health policy

In the aftermath of the 2014–2016 Ebola epidemic in West Africa, the Government of Guinea, with support of development partners made significant commitments and efforts to strengthen the national health system including the improvement of financing, human resources for health, and governance pillars [28–30]. During this period, the National Community Health Policy (politique nationale de la santé communautaire (PNSC)) was adopted in 2017 to improve access and coverage of rural communities to essential health services [31], and in 2018, the PNSC began pilot implementation in 40 convergence communes (municipalities). The adoption of this policy occurred concurrently with other decentralization reforms [29, 32, 33]. In May 2017, the National Assembly adopted the revised code of local authorities which transferred up to 14 competencies to municipalities (Law AN 017), thereby strengthening the role of decentralized levels to manage and implement health services, including for community health. This law aligns with the national decentralization policy (NDP) which aims to contribute to the rebuilding of the State, improving the services provided to the population and, ensuring citizen participation in local public life governance [34]. The NDP is implemented in decentralized communes through the Local Development Plan, which is developed by each commune as the reference planning and budgeting document.

While concurrent national initiatives can promote synergy among health sector reforms, since they are separate vertical lines of administration they may also result in less coordination, especially for complex interventions such as community health programs. It is possible that conflict and confusion of roles and responsibilities generated during the concurrent implementation of such interconnected initiatives may ensue. Substantial gaps between such programs' conceptualization and implementation have been documented in the literature, partly because of poor coordination, power differentials and lack of training, and resources at local level [15, 21]. These challenges point to issues with decentralization and the transfer of power, skills, and responsibilities to stakeholders at lower levels of the health system when implementing complex systems changes such as the community health policy.

While Guinea's Ministry of Health (MoH) has defined the roles and responsibilities of each stakeholder involved in the implementation of the PNSC (**Table 1**), there is significant overlap in roles and responsibilities, and there is early evidence that implementation on the ground diverges substantially from the roles as defined, contributing to implementation challenges [35]. Research indicates that health workers at decentralized levels, such as CHWs, are more likely to succeed when their roles and responsibilities are clearly defined [15]. There is a need to better understand the enabling environment and potential barriers to the effective

**Table 1. De jure roles and responsibilities of stakeholders involved in the implementation of the community health policy.**

| Stakeholders | Roles and responsibilities |
|---|---|
| Ministerial departments (Territorial Administration & Decentralization or MATD; Finances; Budget; Agriculture, Vocational schools; Social Action; and Mining) | • Mobilize material and financial resources for the implementation of the policy<br>• Ensure the monitoring and evaluation of community health activities<br>• Ensure resource transfer to local communities through National agency for financing of communities or Agence Nationale de financement des collectivites (ANAFIC) (National agency for financing of communities)<br>• Control the use of resources |
| Ministry of Public Service and Administration Reform or Ministère de Fonction Publique et de la Réforme de l'Administration (MFPREMA), MATD, MoH | • Ensure compliance with the standards and procedures for career management of local civil servants<br>• Ensure the establishment of the local civil service |
| Livestock/ Environment, Health | • Estimate the number of CHWs, or Agent de Santé communuataires (ASC) and Relais Communautaires, or community organizer (RECOs) per municipality according to the population in collaboration with the Ministry of Territorial Administration & Decentralization<br>• Assess the applications for the recruitment of CHWs in collaboration with the regional level<br>• Check the conformity of the validated lists of ASCs and RECOs sent by the municipalities via the health districts<br>• Supervise the training of ASCs and RECOs<br>• Ensure the application of the national community health policy at the regional level<br>• Organize advocacy meetings for community health financing<br>• Organize multisectoral community health coordination meetings<br>• Support for the implementation of community-based epidemiological surveillance<br>• Support the promotion of community-led total sanitation, environmental hygiene, access, and use of drinking water. |
| Regional authorities | • Participate in the mobilization of material and financial resources for community health<br>• Promote community health<br>• Provide financial and material support for the implementation of interventions<br>• Ensure the coordination of RECOs and ASCs recruitment process in collaboration with Regional Service of Support, Control and Coordination of NGOs or Service régional d'appui, de contrôle et de coordination des ONG (SERACCO)<br>• Supervise the training and post-training of ASCs and RECOs<br>• Ensure the application of the national community health policy at the prefectural level. |
| Prefectural (district level) authorities | • Disseminate legal texts<br>• Supervise recruitment of and participate in the official installation of ASCs and RECOs<br>• Mobilize communities and promote ASCs and RECOs<br>• Ensure the monitoring and evaluation of community health activities<br>• Ensure the application of the Minimum package of activities or Paquet minimum d'activités (PMA) of health centers and posts<br>• Serve as advisory support to municipalities in the identification of health issues, planning, implementation, and evaluation of local health programs<br>• Monitoring (or follow-up) activities of health centers<br>• Ensure the application of the national community health policy at the local level<br>• Integrate health posts into the implementation of major activities such as vaccination and antenatal consultations<br>• Provide training to ASCs and RECOs in collaboration with the head of health center and MATD |
| Mayors | • Mobilize material and financial resources to motivate ASCs / RECOs<br>• Promote income-generating activities<br>• Participate in the official installation of ASCs and RECOs<br>• Mobilize the community<br>• Recruit ASCs and RECOs<br>• Validate the recruitment of ASC/RECO by an order of the mayor<br>• Ensure the monitoring of ASC/RECO activities within the framework of community health promotion |

(*Continued*)

**Table 1.** (Continued)

| Stakeholders | Roles and responsibilities |
| --- | --- |
| Health and Hygiene Committees | • Mobilize communities<br>• Contribute to the mobilization of local resources necessary for the implementation of community-based interventions and motivation of RECOs<br>• Participate in the selection of RECOs<br>• Participate in coordination meetings organized by the managers of health centers and health posts |
| Village Health Committees | • Mobilize communities<br>• Mobilize local resources<br>• Participate in the RECO selection process<br>• Participate in coordination meetings organized by managers of health centers and health posts<br>• Participate in the monitoring of community activities |
| Non-governmental organizations (NGOs) / international / National and local associations | • Mobilize communities<br>• Contribute to resource mobilization<br>• Support the RECO selection process<br>• Contribute to the organization of joint monitoring and evaluation missions for community health activities<br>• Provide the necessary technical, material, and financial support CHWs' activities in collaboration with health and administrative authorities<br>• Report community activities to the health center manager<br>• Participate in coordination meetings organized by the Health District team<br>• Integrate their work plans into the health district plan |
| Technical and financial partners | • Provide technical and financial support<br>• Participate in monitoring and evaluation<br>• Participate in the development of the implementation process |
| Traditional medicine practitioners | • Contribute to the mobilization of communities<br>• Conduct awareness raising activities (Information–Education–Communication (IEC)/ Community-based services, or Service à Base Communautaire (SBC))<br>• Refer/guide cases<br>• Participate in follow-up meetings of interventions at the community level |
| Opinion leaders (parliamentarians, traditional leaders, religious leaders, leaders of associations or groups) | • Mobilize communities<br>• Participate in the mobilization of resources<br>• Participate in the selection process of RECOs<br>• Contribute to the motivation and recognition of ASCs and RECOs<br>• Participate in follow-up meetings of interventions at the community level. |

implementation of the PNSC, along with the reality of how key actors in the health system are currently filling their roles, responsibilities, and decision-making power.

However, there has not been any implementation research to date specifically examining the intersection of these two complex health systems changes: community health and decentralization programs in Guinea. Nearly four years into the launch of Guinea's PNSC, the Directorate of Community Health and Traditional Medicine (DNSCMT) and its partners recognize that this is an opportune moment to investigate the degree to which actors at decentralized levels of the health system understand and are exercising their new roles and responsibilities for delivering community health services, the alignment of these new responsibilities with available resources and capacities, how these factors affect the institutionalization of community health in the context of decentralization, and whether there is early evidence of expected performance outcomes from local actors' and CHWs' actions in relation to routine MCH indicators.

As the implementation of the PNSC has been scaled up to over half of the country (75% or 257 out of 342 communes), the DNSCMT, MoH, and other key stakeholders are considering how to make iterative improvements to the existing policy and process of scale-up. There have been early efforts to assess the implementation of the PNSC. For example, an evaluation in two

regions found a marked increase in the proportion of caregivers that trusted CHWs to treat sick children, in the proportion of children 12–23 months who had received vaccines and, the proportion of pregnant women that received a post-delivery home visit from CHWs in areas of policy implementation. However, attendance of 4 or more ANC visits was still very low in these districts and persisting challenges included demotivating factors such as low CHW salaries and lack of means of transportation [35]. The evaluation highlighted several implementation challenges, including the need for sustainable financing for communes, weak local governance, and poor transfer of skills and capacity building at decentralized levels [35]. Further, an evaluation of the USAID-supported Health Services Delivery activity in Guinea revealed that CHWs are providing services that are inconsistent and variable by region, a reflection of different donor priorities, trainings, and resources by region [36]. As Guinea moves to scale-up the PNSC to the rest of the country and to revise the policy for its implementation in urban and peri-urban areas, it is critical to explore challenges affecting its implementation and to explore why, how and under what circumstances the policy is being implemented according to its design.

To help key community health policy decision makers better understand the barriers and opportunities for more effective implementation of community health programs in Guinea and other contexts, and to provide evidence-based recommendations for improving program efficacy, implementation research using a sequential explanatory mixed-methods design will be carried out.

### Study goal and research objectives and questions

The overall goal of this implementation research is to explore the rollout of the community health policy in Guinea in the context of decentralization, and the role of decision space in explaining gaps between the policy's conceptualization and actual implementation through the following two research questions:

1. To what extent do local public actors know their roles and responsibilities under the community health policy? What factors enable or hinder their ability to carry out these roles/responsibilities?

2. To what extent is the planned, integrated provision of services by CHWs to meet population health needs actually being provided at the community level? How well does this provision of services align with the community's perception of alignment with their needs?

The specific research objectives are:

1. Evaluate the level of knowledge of local public actors of their roles and responsibilities in the implementation of the policy, including the factors that facilitate or hinder the awareness of roles and responsibilities;

2. Identify the factors enabling or hindering the capacity of local public actors to assume these roles/responsibilities, including the decision-making space of the actors involved in different levels of the health system in the deployment of the policy;

3. Analyze how the national decentralization policy has impacted the implementation and the results of the national community health policy;

4. Describe stakeholder perceptions on the effectiveness of all integrated services offered by CHWs to meet the health needs of the population at the community level in Guinea, including understanding whether these perceptions vary by gender and age (young adults aged 18 to 25 vs. older adults); and

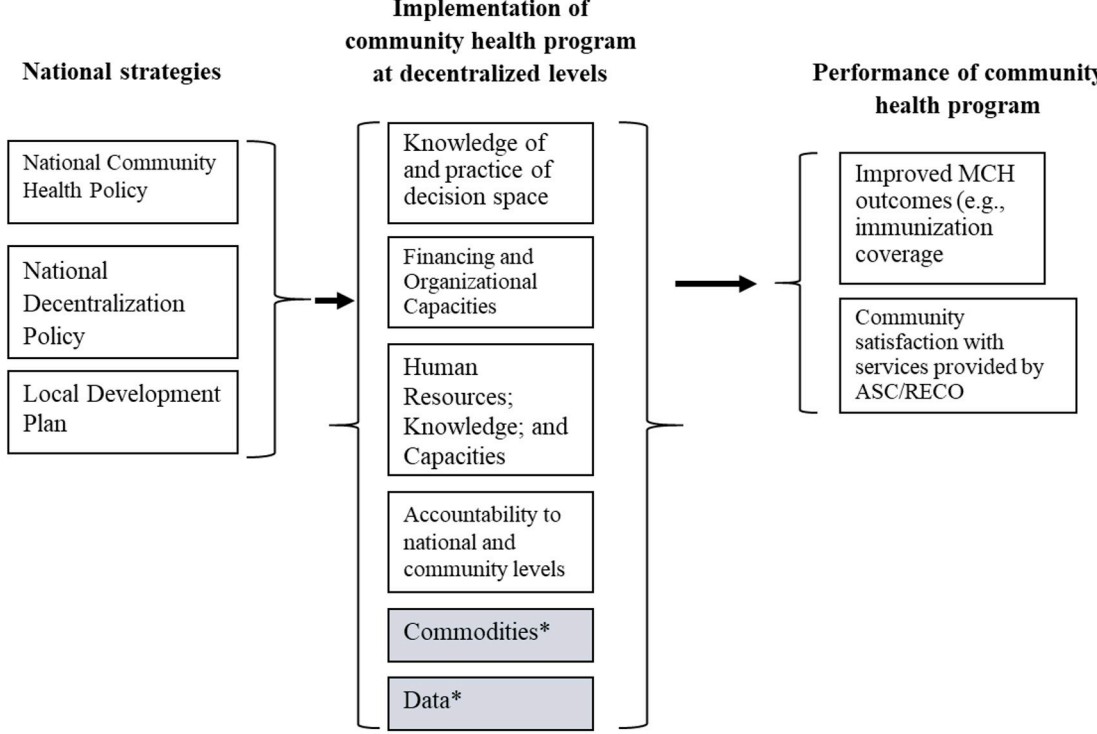

**Fig 1. Theory of change.**

5. Describe any changes that have been observed in key health services (such as the use of the MCH services) since the implementation of the policy.

The theory of change in **Fig 1** describes how we expect the PNSC and associated decentralization reforms in Guinea to improve community health in Guinea, including MCH outcomes (immunization coverage for mother and child, antenatal care coverage and institutional delivery), and community satisfaction with services provided by ASCs and RECOs. We expect that the PNSC and decentralization strategies will achieve improved community health program performance by improving decision space, capacities, and accountability.

Improvements in decision space, capacities, and accountability can be measured via increased knowledge and practice of decision space, increased financial and organizational capacities, increased qualified and trained human resources, increased knowledge and capacities of stakeholders, and improved accountability. Other important factors for these policies and strategies to be successful in their impact on the performance of community health programs include the availability of commodities and the availability of reliable, quality data, although this research study will not be assessing these two factors.

## Materials and methods

### Study design

The implementation research study will employ a sequential explanatory mixed-methods design including the use of quantitative methods and qualitative methods. Mixed methods combine the strengths of both qualitative and quantitative research methods, facilitating

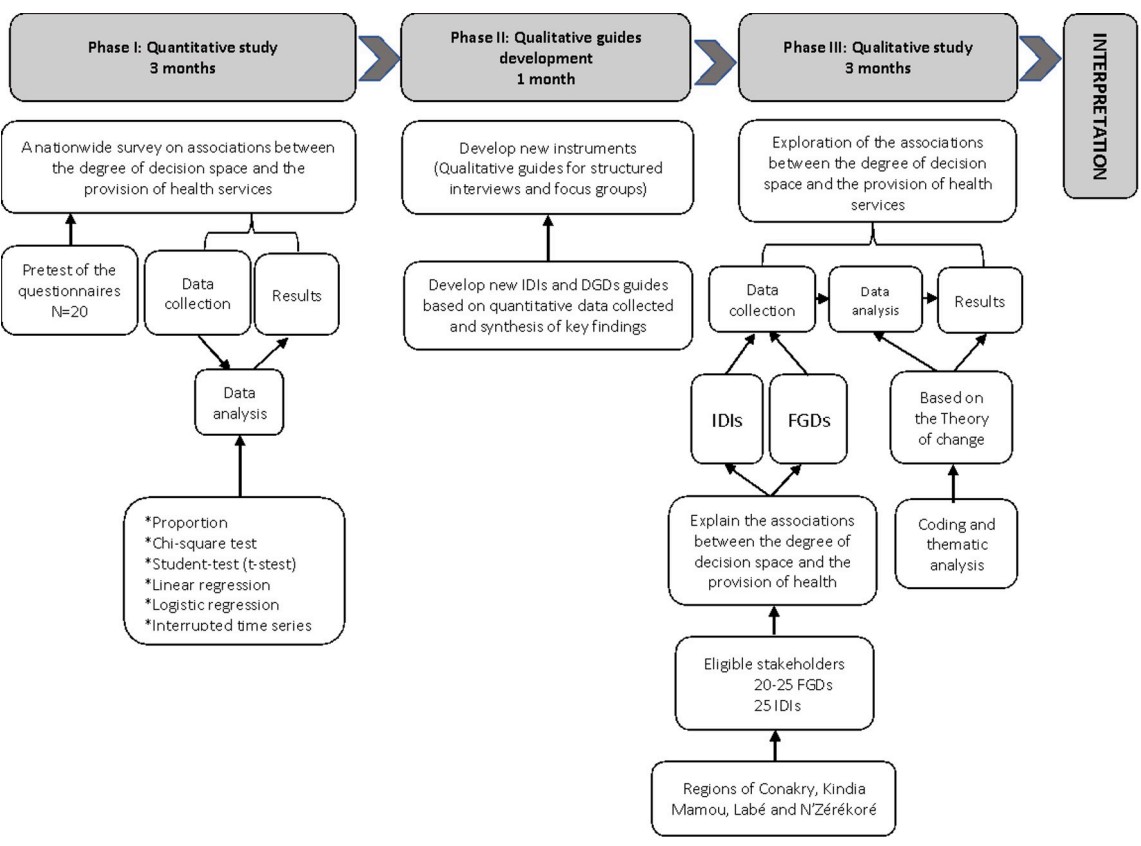

**Fig 2. Decision space sequential mixed study design.**

examination of differences and depth of perspective on the same topic while ensuring flexibility [37]. The sequential design will allow the qualitative data to be used to help explain and further explore the initial quantitative results [38].

The mixed-methods design will use a realist evaluation approach to explain the contextual factors that enabled or hindered the implementation of the policy [39]. The quantitative component, based on a survey questionnaire, will aim at analyzing the decision space among stakeholders and assessing how the policy may be affecting services. Using the quantitative survey findings, we will explore associations between the degree of decision space and the provision of health services using routine service delivery data. The qualitative component will be based on structured interviews with key informants and focus groups. The focus group discussions (FGDs) will provide further insight on the process of implementation of the policy [40]. The sequential study design is described in **Fig 2**.

We will explore the role of decision space, capacities, and accountability in affecting the performance of the national community health policy in Guinea. Performance of the community health program will be assessed via routine MCH indicators and through assessing perceived community member satisfaction with services provided by CHWs.

The research team will begin data collection with the quantitative phase, followed by the qualitative phase involving detailed exploration of the quantitative results and generation of new findings. This sequence is designed to have the qualitative phase help explain the results of the quantitative findings and to explore additional questions that emerge from the quantitative phase.

This study protocol is approved by the National Ethics Committee for Health Research in Guinea (approval date: 29 October 2021, N˚161/CNERS/21) and the Office of Regulatory Affairs and Research compliance of the Harvard TH Chan School of Public Health (approval date: 01, November 2021, IRB21-1162).

## Study context

This study is a collaboration among the Health Systems Strengthening Accelerator project, funded by USAID and implemented by Results for Development, the Maferinyah Training and Research Center in Rural Health, the African Center for Excellence at the Gamal Abdel Nasser University, the DNSCMT of Guinea's MoH, and the Harvard T.H. Chan School of Public Health. The Accelerator project provides ongoing technical assistance to the DNSCMT to support the implementation of the PNSC.

## Research setting

Guinea is characterized by a critical shortage of health care workers (HCWs), with only 7.3 health professionals per 10,000 inhabitants in 2014 [41]. This ratio is three times lower than the threshold recommended by World Health Organization (WHO) in 2006 for the provision of essential facility-based services (23 HWs per 10,000 inhabitants) [5]. This situation is of particular concern for the provision of MCH services in the country, especially in rural areas where 70% of Guineans live, but where only 30% of the health workforce is found [42]. In 2016, only 44% of nurses and 18% of midwives required for MCH were available, contributing to rural-urban disparities in MCH indicators [43, 44]. For example, in 2018, 40% of institutional deliveries occurred in rural areas compared with 84% in urban settings and full immunization coverage of under-five in rural areas was estimated at 21% in 2018 compared with 31% in urban areas [45].

The PNSC, adopted to address these persistent health systems challenges, prescribes an integrated set of essential prevention and care services provided at the community level by two types of community health workers (CHWs): RECO (Relais Communautaire, or community organizer) and ASC (Agent de Santé Communautaire, or Community Health Worker). The RECO provides health promotion, diseases surveillance, and prevention services while the ASC provides a package of basic health services including maternal and child health services [46]. The policy includes standard guidelines on CHW roles, responsibilities, and supervision (details in **Table 2**) [46].

The pilot program launched in 2018 has been extended to other communes. The policy aims to cover all the rural communes in Guinea. It is expected that to achieve full coverage, 18,938 RECO and 1,895 ASC will be recruited and deployed in 338 communes out of 365 total communes in Guinea. Currently the policy is fully implemented and functioning in 69 communes, including the 40 communes of convergence, and 29 other communes. In addition, the policy was supposed to be introduced in 120 additional rural communes (2020–2021) but there have been delays in training and equipping the ASC and RECO. Finally, in 149 communes, the policy has not been introduced at all but some of these communes may have kept the CHWs that existed before the policy was adopted in 2017. However, because they do not meet the criteria set by the MoH (including level of CHW training and rate per population), these communes are classified as not implementing the policy.

For the purpose of this implementation research, communes where the policy is planned to be rolled out but not yet functional (n = 120) and those communes where the policy is not yet introduced (n = 149) will be put in the same category (non-functional communes). Box 1 below provides an overview of commune classification.

Table 2. Profile and roles of community health workers.

| Characteristics | RECO | ASC |
|---|---|---|
| Profile/ selection criteria | • A volunteer willing to serve in his community, resident member having the confidence of the community for his moral probity and availability, in good health to perform the activities, aged between 18 and 50 years, able to read and write in French and to speak the local language | • Assistant-nurses by training (ordinary-level exam + 3 years of medical training), available and accepted by the community, with moral integrity, good command of the local language and able to take responsibility for all activities related to his/her duties. |
| Ratio | • 1 per 650 inhabitants in rural areas<br>• 1 per 1000 inhabitants in urban areas | • 1 ASC (Community health worker) supervises 10 RECO (community relays) |
| Tasks | • Health promotion (information, education, and communication for behavior change)<br>• Disease surveillance and prevention activities | • Enhance the utilization of health services<br>• Health promotion and education with community<br>• Community-based surveillance of epidemics-prone diseases<br>• Provision of basic mother and child healthcare services |
| Temporary list of approved medicines/medical products that CHWs can use | Data reporting sheets/forms | Contraceptives including condoms, Misoprostol, Amoxicillin dispersible 250 mg, oral rehydration solution / Zinc, Ivermectin, Rapid Diagnosis Tests (RDTs), Mebendazole<br>Artemisinin Combination Therapy for adult, adolescent, and infant, Tetracycline ointment, Chlorhexidine 4% gel, Gloves, Safety box, Chlorine and Paracetamol tablet |
| Salary/incentive | 450,000 GNF per month | 1,200,000 GNF per month |

## Box 1. Classification and definition of three commune types in the research study

**Convergence communes** represent 40 municipalities that are benefiting from the transfer, through the National Support Program for Convergence Communes and Community Health and the revised code of local authorities' decentralization process, of 14 competencies including health, education, and public health. These communes are fully implementing the national community health policy according to MoH criteria, with additional support from the Ministry of Administration of Territories and Decentralization.

**Partially functional communes** represent 29 municipalities where the national community health policy is being fully implemented according to MoH criteria with support from development partners such as Global Fund and others. These are designated as partially functional because the decentralization process that has been implemented in the communes de convergence has not been rolled out yet in these communes.

**Non-functional communes** include two categories of communes (269 in total): Municipalities (120) that have been targeted or allocated to development partners to implement the community health policy (i.e. donor funding or commitment has been obtained) but where the ASC/RECO are not yet operational. In these communes, either the ASC/RECO have not yet been recruited or have been recruited but not yet trained and equipped. Non-targeted communes (149): municipalities which have not yet from policy implementation or have not been targeted for support. They may have ASC/RECO with inadequate proportion (ratio per population) or with a profile and training packages different from the requirements of the policy. These communes have also not received donor or government funding to implement the policy.

## Mixed-methods evaluation

### Quantitative component

Both primary and secondary data collection will be gathered.

## a. Primary data

### Survey with decision-makers

*Survey design.* A cross-sectional survey with officials (decision-makers) across national, regional, and local levels will be conducted.

The survey with the officials will be developed to explore the official (*de jure*) and actual (*de facto*) decision space, institutional capacities, and accountability of decision-makers across various levels of health and political systems in Guinea, including in different types of communes, and local officials' level of knowledge or awareness of their roles and responsibilities. The questionnaire was developed by researchers with experience in survey design for decision space analysis as well as researchers and officials with experience at the national, regional and commune levels in Guinea. It will be pre-tested with health providers in Conakry. The final survey will be programmed in KoBoToolbox and uploaded into tablets.

*Survey population.* The survey will be conducted with stakeholders involved in the implementation of the PNSC at the community, district, regional and national levels of the health system. The profile of survey respondents is summarized in S1 Table. National level stakeholders are only minimally targeted by the survey because they fit more with the qualitative component where in-depth interviews are more suitable to capture a wealth of information on the policy design and rollout

*Sampling.* A two-level stratified sampling method will be used; the two levels are the administrative region and commune levels. Information on the status of the community health policy by the time of data collection will be used to guide this selection process.

At the first stage, we proceeded with purposive sampling of four administrative regions of the country after discussion with MoH (DNSCMT) in a way to ensure good representativeness of different regional contexts (see S2 Table). S2 Table shows all the regions in Guinea except for Conakry, which is excluded because it is an urban area, and this policy is being rolled out in rural regions. We categorized regions that had similar demographic and health indicators (for instance, rural population, adult literacy, effective immunization, and anemia in children). After discussion with the leadership of the MoH (DNSCMT), the following four study regions were purposely selected to reflect variation in the implementation status of the community health strategy: Kindia, Mamou, Labé and N'Zérékoré. For example, Kankan and N'Zérékoré regions had similar indicators and, therefore, N'Zérékoré was included. S3 Table describes the level of the community health strategy implementation by commune and by region. S4 Table describes the variables and data sources for the primary data collection.

At the second level, we will randomly select two districts (prefectures) per region from the list of districts in that region. We will then randomly select seven communes in the two selected districts in Kindia, Mamou, and N'Zérékoré (four communes in one district and three in the other district), and six communes in Labé region (three per district). The communes will be selected independently of their category (convergence communes, functional communes, and non-functional communes) using random selection function in MS Excel 365. (see S3 Table).

At the commune level, all participants of a given category will be surveyed, except for RECOs for whom a purposive sampling will be used to select up to five of them per commune.

The quantitative study sample was adjusted after ethics approval because the new government in Guinea proceeded with new nominations at all levels of the health system. Therefore, where the director or head at the regional, prefectural, and commune levels may have changed, we will plan to interview the deputy to account for the new administrative context. Also, as some previous health system directors might be transferred within areas where the study is conducted, they will be interviewed about their previous position.

*Data collection.* Data collectors (14 in total) and their supervisors will be trained during a three-day workshop to ensure they are all familiar with implementation research procedures and tools (see S1 Fig). They will be qualified data collectors (medical doctors, sociologists, master's in public health students or community health specialists). The supervision team will include implementation research staff and senior staff from the DNSCMT. Standard preventive measures (masking, social distancing, etc.) will be used to protect the research team and participants against Coronavirus disease 2019 (COVID-19) transmission.

*Data analysis.* The analysis will be carried out using Stata software (version 16.1). Descriptive statistics will be generated for independent variables using proportions (with 95% confidence intervals) and means (with standard deviation) as applicable. A flexible approach for analysis is envisioned. We will consider carrying out three exploratory factor analyses to assess whether decision space, capacity, and accountability latent constructs can be identified from the measured survey questionnaire items. In this approach, any individual items that do not adequately measure these constructs will be dropped. Then, these dependent variables (degree of decision space; capacity; and level of accountability) will be summarized as mean standardized scores. Further analysis will be conducted to explore findings across the levels of the health system, regions, and type of communes. For the analysis, we will use a linear regression model to assess the variations of dependent variables across other variables. Second, we will explore the use of a logistic regression model to predict high (versus low) levels of decision space, capacity, and accountability.

## b. Secondary data

### Design

Routine data collection at the health center level will be conducted in the selected communes whatever their category is (policy fully, partially, or not yet implemented). We will use a retrospective longitudinal survey design.

### Data sources and collection

Data covering 12 months prior to ASC/RECO activities start up to March 2021 will be collected. Particular emphasis will be placed on the key indicators related to ASC/RECO' mandate. These include MCH indicators such as antenatal, postnatal and outpatients' visits, childbirth, and immunization coverage.

Data will be validated across sources at the health facility, including logbooks, facility medical records, and health cards, in order to check data quality and consistency. S5 Table describes the variables and data sources for the secondary data collection.

### Data analysis

The research team will explore the potential results of the policy through comparative analysis between areas where the policy is fully and partially implemented and where it is not implemented, and between communes of convergence and ordinary communes.

First, we will use the Chi square test and Student's t-test to assess individual relationships between each outcome and the independent variables. We will use the Chi-Square test to compare MCH coverage indicators according to the type of commune and per region and Student's t-test for comparing the annual mean numbers of each dependent variable (i.e., immunization coverage, antenatal care, assisted deliveries, new users of family planning) before and during the community health policy intervention, across the types of communes, and between implementation and comparison communes.

Next, we will use ordinary least squares (OLS) multiple regression to compare MCH coverage indicators according to the type of commune and per region controlling for the covariates in S5 Table. OLS multiple regression will be used to compare the dependent variables across the types of communes and between implementation and comparison communes controlling for the covariates in S5 Table. Other key indicators of the community health strategy include the numbers of children born with a birth certificate, receiving Penta 3, fully vaccinated, sleeping under LLINs (long-lasting insecticidal nets), and screened for malnutrition. We will create categorical variables (low, medium, high) for decision space, accountability, and capacity for analyses pertaining to routine data. We will carry out OLS multiple regression to assess the association between decision space, accountability, and capacity and the key MCH indicators of interest controlling for the covariates in S5 Table. For all analyses, the level of significance will be set at $p = 0.05$.

For the final step of the quantitative analysis, we will carry out an interrupted time series analysis using facilities' routine data to assess changes in routine MCH service delivery indicators (outcome variables) that are linked to ASC/RECO activities from the year prior to the community health policy implementation through December 2021. Single and multiple groups analyses will be used to measure changes in both levels and trends after policy implementation in the intervention group. Monthly aggregated data for each outcome will be considered as the points in the sequence of observations. The general Cumby-Huizinga error autocorrelation test (actest) will be performed to assess the general series specification of each model. The null hypothesis of this test is that there is no autocorrelation in the error distribution. If autocorrelation is present, the lag level will be specified accordingly before the final model is selected. The validity of each model will be assessed using the overall p-value. Specifically, for the comparison models (multiple groups), we will make sure that trends of the outcome are similar between the intervention and control groups in the pre-intervention period before we continue the analyses. All statistical analyses will be performed at a 5% threshold, and differences in levels and trends will be considered statistically significant at $p\text{-value} < 0.05$.

## Qualitative component

Once the results of the quantitative component are available and analyzed, the qualitative data collection tools will be developed, validated by MoH and USAID, pilot-tested and qualitative data collection will be conducted. As the qualitative components of this research (In-depth interviews (IDIs) and Focus group discussions (FGDs)) will be developed and launched sequentially with the quantitative survey, they will not only explore further the quantitative findings and hypotheses developed by the research team but will also provide the opportunity to delve further into explaining unexpected initial findings from the quantitative survey, as well as illuminate new, additional findings. The qualitative component will include a maximum of 25 IDIs (in rural communes) and 20–25 total FGDs, including key informants from local, district, regional, and national levels (see S6 Table). These numbers reflect the maximum that this study will include, but final numbers and selection of participants will be determined based on the results of the quantitative survey, and the areas in which we are interested in

probing and exploring more through IDIs and FGDs. The proposed number of IDIs and FGDs is based on the assumption that these numbers will be sufficient to reach the saturation point of data collection. We will determine that saturation has been reached when additional IDIs and FGDs do not reveal new themes, insights, or information useful to address they study's objectives, and when the issues raised in the quantitative analysis for further qualitative research have been adequately explored.

We will conduct separate FGDs at each level of the health system to triangulate and further explain and understand the quantitative findings. The qualitative component will target officials across the health and political systems to understand why for instance "de jure" responsibilities do not match with "de facto" ones, and what mitigating layers are in place, and where. The implementation research will explore to what extent local officials are involved in accountable and transparent decision-making processes in the policy implementation.

Community representatives, grassroots organizations, and beneficiaries of health services will also be interviewed to better understand the effect of the national community health policy rollout on their access to and utilization of health services. The qualitative data will be collected at the commune, regional and national levels.

All qualitative guides will be developed in French and, where needed, translated in local languages during interviews. After obtaining informed consent, interviews will be recorded.

### Data collection

The qualitative teams will consist of ten data collectors (five per team) in charge of conducting IDIs and FGDs and developing a reflexive summary after each interview or group discussion. Each team will be supervised by a qualitative data manager in charge of reviewing the reflexive summaries, transcription of IDIs and FGDs and data coding in NVivo software. They will also ensure good quality of data collection through rigorous follow-up with data collectors during the data collection process and assist in the elaboration of the qualitative data analysis plan.

### a. Focus group discussions (FGD)

Each FGD will have a total of 6 to 8 participants and will be scheduled to last for an estimated 60 minutes. It is likely that FGDs will not be evenly spread across all communes, but rather will be purposefully selected based on the findings from the quantitative data. The FGDs will be used as a tool to dive more deeply into questions that arise based on the analysis, and the regions, types of communes, and participants of the FGDs will be determined based on findings, outliers, and anomalies the research team identifies in the quantitative findings. These FGDs could include health care workers, ASC/RECO, and community (including members of health and hygiene committees, and representatives from community-based organizations), as well as district, regional actors in the health sector and other sectors. Depending on the findings from the quantitative data collection, focus group discussions may also include national level actors and donors/partners to gain more details on their views and perspectives about the past, present, and future of the community health strategy.

Specific focus group guides will be developed by the research team and will integrate questions related to the implementation and sustainability of the community health strategy.

### b. In-depth interviews (IDI)

Participants of the IDIs will be chosen based on the principles of flexibility and maximum variation [40]. These interviews will enable collection of detailed and nuanced information beyond the quantitative survey and FGD.

The IDIs will be conducted using a stakeholder-specific interview guide that integrates each of the implementation research questions relevant to the type of official. Each IDI will be scheduled to last an estimated 60 minutes.

We anticipate that three IDIs will be sufficient by target commune (i.e., the mayor, the sous-préfet and one religious leader or member of a local NGO). We do not anticipate conducting IDIs in districts and regions but depending on the findings from the quantitative component, the final number of IDIs will be decided upon.

### Data analysis

The recorded IDIs and FGDs will be transcribed verbatim into French (irrespective of the language of the interview), and then coded using NVivo software. We will then analyze the transcripts using thematic analysis using a codebook. We will compare the data from the various sources to triangulate the data and thus strengthen the internal validity (credibility) of the study [47]. Inter-coding bias will be addressed by having two qualitative experts coding the material for intercoder reliability [47].

### Ethic statement

This is an implementation research activity which results may be used by the Ministry of Health and R4D for decision making and programming. This protocol was approved by the Guinea National Ethics Committee for Health Research in Guinea (CNERS) and the Harvard T.H. Chan School of Public Health for approval.

All IDIs and focus group discussions will take place in private locations. Names will not be associated with notes or other study materials. Free and informed consent will be obtained verbally from the participants (given the context of EVD and COVID-19 epidemics, the written consent increases the risk of contamination for both the participant and the enumerators). However, it will be documented electronically on data collection tablets. All recordings made during interviews or photos taken for the purposes of the implementation research report will require a completed consent form. The informed consent process will include communication to participants that the anonymized data may be submitted to USAID and used for publication.

## Discussion

This study will evaluate the implementation of the PNSC and the National Decentralization Policy and these policies' contributions to improving: (1) MCH service delivery indicators including immunization coverage for mother and child, antenatal care coverage, and institutional delivery; (2) community satisfaction with services provided by community health workers recruited through these new policies and strategies; (3) and the performance of community health programs in a decentralized setting including local health system actors' decision space, capacities, and accountability [27].

### Study strengths

The strengths of this study include its robust, theory-informed, mixed implementation research design and its participatory approach involving the Guinea MoH and other stakeholders in the design, collection, verification, and validation of implementation research findings. This approach will help ensure that the research is practical and will maximize the changes that the findings will be directly used by the MoH and its partners. Furthermore, it considers the context of policy initiation and implementation, the policy's current implementation status, and future prospects.

### Potential limitations of the study

Limitations of this study include the quality of routine data; potential confounds or omitted variable bias; social desirability bias; potential inaccessibility of study sites due to COVID-19 and other safety concerns; and external generalizability to regions not included in the study. Each limitation and mitigating measures are discussed below.

We will use routine data from health center levels and DHIS2 (District Health Information System II). The quality of both sources can be a potential limitation, given the current difficulties in receiving timely, precise, and reliable data. To minimize these biases, we will triangulate the data from different sources available at the health facility including logbooks, facility medical records, and health cards to check data quality and consistency. Further, other confounding factors may be responsible for observed differences in service delivery indicators and or decision space across geographic regions. We will try to control for some potential confounders by including covariates described above (health indicators, population size, etc.) and by collecting these data up to 12 months prior to the policy rollout to account for pre-existing differences using interrupted time series analysis. However, there is still the potential for omitted variable bias in driving observed differences.

Participants in the quantitative or qualitative data collection components may be influenced to give responses that they perceive to be desired by the research team contributing to social desirability bias. This will be mitigated to the extent possible through the informed consent process (making it clear that participation does not impact any ongoing support received for program implementation and that participation is completely voluntary) and by in-depth training of data collectors to avoid asking leading questions and to pay attention to body language, facial expressions, etc.

In the current context of COVID-19 and other epidemics in Guinea, access to all targeted stakeholders and study sites may be constrained. This will be mitigated as much as possible by secondary document review and virtual interviews when in person data collection is not feasible. Moreover, the outcomes of the community health program may not be generalizable to all health facilities and services in Guinea, which means that the external validity of the results may be limited. To mitigate this concern, the four implementation research regions have been selected in consultation with the DNSCMT/MoH to reflect maximum variation in health profiles, sociodemographic status, and policy implementation.

### Expected outcomes of the study, public engagement, and dissemination

The study is designed to have direct, practical influence on the PNSC implementation. Preliminary results will be shared with the MoH, community health stakeholders, and USAID for internal analysis and feedback. Virtual meetings will be held to discuss, synthesize, and validate the findings. This will lead to the formulation of study recommendations. The findings from this report will be presented during a dissemination event for key stakeholders. Study findings will also be disseminated through policy briefs in Guinea, through national and international scientific meetings, and peer-reviewed publications.

## Conclusion

This study will contribute to the understanding of factors that influence the implementation of community health programs in a context of decentralization process in Africa, and the role of decision space in explaining gaps between the policy's conceptualization and actual implementation.

The findings and recommendations will inform the iterative design and improvement for the scale-up of the PNSC and can help to guide decision-makers in the health sector and

beyond on the need to readapt the program for its improved efficacy, including any policy changes or revisions to improve community health program governance and implementation and refined or clarified roles and responsibilities for actors responsible for the community health program. The study findings can also be used to target capacity-strengthening efforts to actors at different levels of the health systems and in different implementation regions. Gaps between the policies design and implementation can highlight bottom-up and top-down accountability issues for improving responsiveness of the community health program to communities' needs.

## Supporting information

**S1 Fig. Study timeline.**
(TIF)

**S1 Table. Survey target population.** National level stakeholders are only minimally targeted by the survey because they fit more with the qualitative component where in-depth interviews are more suitable to capture a wealth of information on the policy design and rollout.
(DOCX)

**S2 Table. Selected population and health indicators by region in Guinea.** Regions selected based on demographic and health indicators.
(DOCX)

**S3 Table. Estimated sample of communes by level of community health policy implementation and by region in Guinea.**
(DOCX)

**S4 Table. Variables and data sources for primary data.**
(DOCX)

**S5 Table. Variables and data sources for secondary data.**
(DOCX)

**S6 Table. Qualitative study target population and sample.**
(DOCX)

## Author Contributions

**Conceptualization:** Alexandre Delamou, Fassou Mathias Grovogui, Lior Miller, Amy Nye, Delphin Kolié, Thomas J. Bossert.

**Writing – original draft:** Alexandre Delamou, Fassou Mathias Grovogui.

**Writing – review & editing:** Lior Miller, Amy Nye, Mamadi Kourouma, Tohanizé Goumou, Thomas J. Bossert.

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
