## [Decision Letter · Decision Letter 0]

4 Aug 2022

PONE-D-22-10462Implementation research protocol on the National Community Health Policy in Guinea: A sequential mixed-methods study using a decision space approachPLOS ONE

Dear Dr. Fassou Mathias Grovogi,

Thank you for submitting your manuscript to PLOS ONE. After careful consideration, we feel that it has merit but does not fully meet PLOS ONE’s publication criteria as it currently stands. Therefore, we invite you to submit a revised version of the manuscript that addresses the points raised during the review process.

We look forward to receiving your revised manuscript.

Kind regards,

Chulaporn Limwattananon, Ph.D.

Academic Editor

PLOS ONE

Journal Requirements:

Reviewers' comments:

Reviewer's Responses to Questions

**Comments to the Author**

1. Does the manuscript provide a valid rationale for the proposed study, with clearly identified and justified research questions?

Reviewer #1: Yes

Reviewer #2: Yes

2. Is the protocol technically sound and planned in a manner that will lead to a meaningful outcome and allow testing the stated hypotheses?

Reviewer #1: Yes

Reviewer #2: Yes

3. Is the methodology feasible and described in sufficient detail to allow the work to be replicable?

Reviewer #1: Yes

Reviewer #2: Yes

4. Have the authors described where all data underlying the findings will be made available when the study is complete?

Reviewer #1: No

Reviewer #2: Yes

5. Is the manuscript presented in an intelligible fashion and written in standard English?

Reviewer #1: Yes

Reviewer #2: Yes

6. Review Comments to the Author

You may also provide optional suggestions and comments to authors that they might find helpful in planning their study.

Reviewer #1: This is a well written protocol. However, I have some concerns about the role of the Government and the funder in the research. As I understand your proposal the research is being funded by USAID and the research is for the Government to inform revisions to the National Community Health Policy to improve its rollout and effectiveness. Two of the authors are employed by the Ministry of Health in Guinea. You need to reflect more carefully how independent the research will be. Do the Government employees have any conflict of interest? Will the research be independent of influence by the funder and/or the Government?

More minor points:

1. you say the RECO are volunteers and then say they have a salary of 450,000GNF per month - are they volunteers or employees?

2. In the qualitative research you say that two IDIs will be interviewed but I think that you list three role?

Reviewer #2: Thank you to the authors for a well written manuscript. This will provide valuable data in decentralisation of community health worker programmes and perhaps ways in which the programme may be strengthened and made sustainable.

7. PLOS authors have the option to publish the peer review history of their article (what does this mean?). If published, this will include your full peer review and any attached files.

Reviewer #1: No

Reviewer #2: No

---

## [Author Response · Author response to Decision Letter 0]

12 Dec 2022

Chulaporn Limwattananon, Ph.D.

Academic Editor

PLOS ONE

Manuscript ref. num: PONE-D-22-10462

 December 07, 2022

Dear Editor,

We are very grateful to the Editor and the reviewer for the comments they provided on our paper entitled “Implementation research protocol on the National Community Health Policy in Guinea: A sequential mixed-methods study using a decision space approach”. We have received two sets of comments from the Editor : (i) August 4th, 2022, and (ii) October 10th, 2022. We have addressed all those comments and organized our responses by date of reception of those comments. We feel that these revisions have significantly improved our protocol content. Therefore, please find our point-by-point response below to each comment from the Academic editor and the reviewers in bold font followed by our responses in italic. We have also updated the manuscript as requested.

Comments received on August 4th, 2022 (page 1 to 5)

RESPONSE TO THE ACADEMIC EDITOR COMMENTS : 

We have formatted the protocol accordingly.

Funding for this study was provided by the United States Agency for International Development (USAID). Funding was provided by USAID to the organization Results for Development (R4D) under the Health Systems Strengthening Accelerator Project under the grant number No. 7200-AA-18CA-00037. The grant information can be found at the following address https://govtribe.com/award/federal-grant-award/cooperative-agreement-7200aa18ca00037. None of the author received a grant received a grand with his/her name. We have addressed it during the submission. This comment has also been addressed in the additional comments received later (October 10). Please see below (page 6, comment 1).

3.Your ethics statement should only appear in the Methods section of your manuscript. If your ethics statement is written in any section besides the Methods, please delete it from any other section.

We confirm that this is already part of the methods section (see lines 492 -504) of the clean copy of the revised version

We have revised the list of references. We originally had 48 and now we just have 47. This is due to an error when referencing the first version submitted to your journal. We have now found that the original 47 reference was not retained in the protocol.

Note: HTML markup is below. Please do not edit.

We did not understand this note 

RESPONSE TO THE REVIEWERS’ COMMENTS

1. Does the manuscript provide a valid rationale for the proposed study, with clearly identified and justified research questions?

Reviewer #1: Yes

Reviewer #2: Yes

We thank the reviewer for this appreciation

2. Is the protocol technically sound and planned in a manner that will lead to a meaningful outcome and allow testing the stated hypotheses?

Reviewer #1: Yes

Reviewer #2: Yes

Thank you for this comment

3. Is the methodology feasible and described in sufficient detail to allow the work to be replicable?

Reviewer #1: Yes

Reviewer #2: Yes

Thank you for this comment

4. Have the authors described where all data underlying the findings will be made available when the study is complete? 

Reviewer #1: No

Reviewer #2: Yes

We agree with the reviewer that the information we provided at this stage was not clear. In fact, no datasets were generated or analyzed during the study design time. All relevant data from this study will be made available upon study completion.

5. Is the manuscript presented in an intelligible fashion and written in standard English?

Reviewer #1: Yes

Reviewer #2: Yes

Thank you for this comment

6. Review Comments to the Author

Reviewer #1: This is a well written protocol. However, I have some concerns about the role of the Government and the funder in the research. As I understand your proposal the research is being funded by USAID and the research is for the Government to inform revisions to the National Community Health Policy to improve its rollout and effectiveness. Two of the authors are employed by the Ministry of Health in Guinea. You need to reflect more carefully how independent the research will be. Do the Government employees have any conflict of interest? Will the research be independent of influence by the funder and/or the Government?

More minor points:

Thank for this remark. This research led by researchers from the University of Conakry who have proven their scientific integrity in conducting similar evaluations in Guinea. We have already confirmed in the cover letter and submitting process that none of the authors has declared competing or conflict of interest. We confirm that this research will be independent of influence by the funder and/or the Government.

1.you say the RECO are volunteers and then say they have a salary of 450,000GNF per month - are they volunteers or employees?

We thank the reviewer for his comments. Yes, we confirm that RECOs are community volunteers who devote some, but not all, of their time to community health activities, unlike CHWs who are recruited and dedicate their time to community health work. RECOs do not receive a salary but monthly incentives.

2.In the qualitative research you say that two IDIs will be interviewed but I think that you list three roles?

We thank the reviewer for this remark. We have addressed it (see line 483).

Reviewer #2: Thank you to the authors for a well written manuscript. This will provide valuable data in decentralisation of community health worker programmes and perhaps ways in which the programme may be strengthened and made sustainable.

We thank the reviewer for his/her kind comments

7. PLOS authors have the option to publish the peer review history of their article (what does this mean?). If published, this will include your full peer review and any attached files.

If you choose “no”, your identity will remain anonymous, but your review may still be made public.

Do you want your identity to be public for this peer review? For information about this choice, including consent withdrawal, please see our Privacy Policy.

Reviewer #1: No

Reviewer #2: No

We respect the reviewers’ choice not to have their identity published

Comments received on October 10th, 2022 (page 6)

1. We note that the grant information you provided in the ‘Funding Information’ and ‘Financial Disclosure’ sections do not match. 

This study was supported by the USAID through a grant (No. 7200-AA-18CA-00037). We have provided the right grand number during the submission. 

"https://www.usaid.gov/

The funder reviewed the protocol before it submission"

Funding for this study was provided by the United States Agency for International Development (USAID). Funding was provided by USAID to the organization Results for Development (R4D) under the Health Systems Strengthening Accelerator Project. 

b) State what role the funders took in the study. If the funders had no role in your study, please state: “The funders had no role in study design, data collection and analysis, decision to publish, or preparation of themanuscript.”

USAID was involved in reviewing, validating, and approving the study design. They reviewed and provided comments on the protocol/manuscript before submission. They also review findings and were consulted for approval in the decision to publish.

See below.

The authors received no specific funding for this work. USAID provided funding to Results for Development (R4D), and employees and consultants of R4D carried out this study.

---

## [Editor Report · Decision Letter 1]

14 Dec 2022

PONE-D-22-10462R1Implementation research protocol on the National Community Health Policy in Guinea: A sequential mixed-methods study using a decision space approachPLOS ONE

Dear Dr. Fassou Mathias Grovogui,

Thank you for submitting your manuscript to PLOS ONE. After careful consideration, we feel that it has merit but does not fully meet PLOS ONE’s publication criteria as it currently stands. Therefore, we invite you to submit a revised version of the manuscript that addresses the points raised during the review process.

Exhaustive lists of abbreviations (e.g., ANC, ASCs, RECOs, SERACCO, MATD, MoH, MOH, PNSC, DNSCMT, IEC/SBC, RDTs, etc) are required. The first time of presenting an abbreviation begin with a full name or word (e.g., ante natal care) followed by corresponding abbreviation of such a word within the parenthesis (e.g., ANC).

Tables 3, 4, 5, 6, 7 and 8 and Figure 3 should be moved into the supplement files so as to make the paper not too long.

Box 1: an abbreviation of Ministry of Health: MoH and MOH are not consistent. This should be either MoH or MOH through out.

We look forward to receiving your revised manuscript.

Kind regards,

Chulaporn Limwattananon, Ph.D.

Academic Editor

PLOS ONE
---

## [Author Response · Author response to Decision Letter 1]

2 Jan 2023

Chulaporn Limwattananon, Ph.D.

Academic Editor

PLOS ONE

Manuscript ref. num: PONE-D-22-10462

 December 30, 2022

Dear Editor,

We are very grateful to the Editor for considering publishing our protocol entitled “Implementation research protocol on the National Community Health Policy in Guinea: A sequential mixed-methods study using a decision space approach” in your esteemed Journal. Thank you for the different comments to which we have provided responses. We feel that these revisions have significantly improved our protocol content. Therefore, please find our point-by-point response below to each comment in bold font followed by our responses in italic. We have also updated the manuscript as requested.

Comments received on August 4th, 2022 (page 1 to 5)

RESPONSE TO THE ACADEMIC EDITOR’S COMMENTS : 

Thank you for submitting your manuscript to PLOS ONE. After careful consideration, we feel that it has merit but does not fully meet PLOS ONE’s publication criteria as it currently stands. Therefore, we invite you to submit a revised version of the manuscript that addresses the points raised during the review process. 

We thank the Editor for this positive remark. We have improved the protocol content.

Exhaustive lists of abbreviations (e.g., ANC, ASCs, RECOs, SERACCO, MATD, MoH, MOH, PNSC, DNSCMT, IEC/SBC, RDTs, etc) are required. The first time of presenting an abbreviation begin with a full name or word (e.g., ante natal care) followed by corresponding abbreviation of such a word within the parenthesis (e.g., ANC).

We have reviewed the protocol and updated the abbreviations with full name 

Tables 3, 4, 5, 6, 7 and 8 and Figure 3 should be moved into the supplement files so as to make the paper not too long.

We have uploaded the supplementary files during resubmission

Box 1: an abbreviation of Ministry of Health: MoH and MOH are not consistent. This should be either MoH or MOH through out.

We thank the Editor for this comment. It should be MoH and we have corrected it in the protocol.

We would like to keep our financial disclosure at it’s current state.

JOURNAL REQUIREMENTS:

We have checked our references and there is no need to replace any of them

---

## [Editor Report · Decision Letter 2]

5 Jan 2023

Implementation research protocol on the National Community Health Policy in Guinea: A sequential mixed-methods study using a decision space approach

PONE-D-22-10462R2

Dear Dr. Fassou Mathias GROVOGUI,

We’re pleased to inform you that your manuscript has been judged scientifically suitable for publication and will be formally accepted for publication once it meets all outstanding technical requirements.

Kind regards,

Chulaporn Limwattananon, Ph.D.

Academic Editor

PLOS ONE
---

## [Editor Report · Acceptance letter]

9 Jan 2023

PONE-D-22-10462R2 

Implementation research protocol on the National Community Health Policy in Guinea: A sequential mixed-methods study using a decision space approach 

Dear Dr. GROVOGUI:

I'm pleased to inform you that your manuscript has been deemed suitable for publication in PLOS ONE. Congratulations! Your manuscript is now with our production department. 

Kind regards, 

on behalf of

Dr. Chulaporn Limwattananon 

Academic Editor

PLOS ONE